# Exploring the Drivers of Sentinel-2-Derived Crop Phenology: The Joint Role of Climate, Soil, and Land Use

Sofia Bajocco [1], Silvia Vanino [1,*], Marco Bascietto [2] and Rosario Napoli [1]

1. Council for Agricultural Research and Economics, Research Centre for Agriculture and Environment (CREA-AA), 00184 Rome, Italy; sofia.bajocco@crea.gov.it (S.B.); rosario.napoli@crea.gov.it (R.N.)
2. Council for Agricultural Research and Economics, Research Centre for Engineering and Agro-Food Processing (CREA-IT), 00015 Monterotondo, Italy; marco.bascietto@crea.gov.it
* Correspondence: silvia.vanino@crea.gov.it

**Abstract:** The exploration of crop seasonality across a region offers a way to help understand the phenological spatial patterns of complex landscapes, like agricultural ones. Knowing the role of environmental factors in influencing crop phenology patterns and processes is a key aspect for understanding the impact of climate and land use changes on agricultural landscape dynamics. We identified pixels with similar phenological behavior (i.e., pheno-clusters) and compared them to the land cover map of the study area to assess the role of the land management component in controlling the phenological patterns identified. Results demonstrated that soil texture is the most important factor for permanent crops, while large amount of rainfall and high values of available water content are the main drivers in spring cultivations (i.e., irrigated crops). Scarce water availability (in terms of soil texture, low annual precipitation and high minimum temperature) represented the main driving factor for non-irrigated crops, whose phenology is characterized by summer drought and fall-winter productivity. Compared to vegetation maps that use only land cover from a single season or period, using seasonality of the NDVI time series to classify the agricultural landscape provides different and more ecologically relevant information about croplands.

**Keywords:** agroecosystem; Copernicus Sentinel-2; Mediterranean; multivariate analysis; phenology

## 1. Introduction

Phenology is the study of the timing of recurring biological events, the causes of their timing, and the related biotic and abiotic driving forces [1]. Accordingly, phenology can contribute to many scientific disciplines, from climate change, biodiversity, agriculture, and forestry to human health. The knowledge of timing of phenological events and their spatial variability can provide valuable data for land-use planning, crop zonation, pest control, species conservation and protection, and pollen release and its implications for human health [2].

The investigation of phenology has a long tradition in agriculture and its long-term interest has come from the need for understanding plant development and growth dynamics and their relation to the surrounding environment. Recently, considerable literature [3,4] has grown up around the topic of crop phenology as an important parameter for crop growth monitoring, yield prediction, growth simulation, and decision-making tools to face climate change.

As the impacts of climate change intensify, the need to understand the functioning of the agro-ecosystems has stimulated scientific communities to elucidate environmental controls on vegetation dynamics [5]. Phenology variables are indicated as some of the most sensitive data to climate conditions, and therefore represent key indicators of crop growth and development and play an important role in vegetation monitoring [6]. However, compared to natural vegetation, crop growth is not only driven by natural (climate and soil) conditions, but also modified through field management activities. Agro-ecosystems are

strongly affected by natural conditions and human activities [4,7,8]. Patterns in crop growth are influenced by processes involving land use type, soil conditions, water availability, and regional climate [8,9], and any changes in crop phenology are closely related to these environmental controlling factors.

Satellite-based observations with a wide spatial coverage and short revisit times have become a valuable tool for monitoring vegetation growth and retrieving vegetation phenology based on remotely sensed vegetation index (VI) time-series data [4,10,11]. Vegetation index time series are powerful indicators reflecting the dynamics of vegetation growth and vegetation coverage, such as the Normalized Difference Vegetation Index (NDVI) [12]. The NDVI is the most used vegetation index applied in agricultural applications and is a measure of photosynthetic capacity of the vegetation cover [13]. The advantage of this method is that vegetation phenological information can be continuously monitored at local to global scales. On the downside, it must be noted that the satellite-based indicators do not directly infer crop development stages (in sensu stricto), but are, instead, monitoring crop growth and the intra-seasonal variations of the agricultural land cover [14], which is not always closely linked to key developmental events [4]. For this reason, the scientific community generally refers to land surface phenology (LSP) when satellite-based techniques are used [3].

The appearance of crop profiles is affected by regional variations in climate, soil and management practices, and satellite images help to interpret crop vitality, soil properties, and climate stress. Hence, using VI time series, information about cropping phenology patterns can be extracted by examining the number of peaks in a vegetation index time series, which corresponds well to the growing cycles of crops, such as heading, maturing, and senescence [8]. For example, vegetation index time series of single cropping presents only one growth cycle per year, while that of double cropping presents two cycles [15]. Hence, cropping phenology patterns of a territory can be identified and mapped by examining the periodic variations in the vegetation index time series [8,16,17]. The crop seasonality should be accounted for by setting-up individual crop profiles for each homogenous agroregion [18,19]. Therefore, quantifying how cropping phenology patterns respond to this environmental forcing at landscape scale is crucial for understanding crop spatio-temporal dynamics [20].

While in forestry the role of the environmental factors in influencing forest phenology patterns and processes at territorial scale has been largely explored [21–23], in agricultural studies this kind of research question has received less attention, maybe due to the more dynamic character of agricultural systems [19] and the major interest towards crop type identification (Gao and Zhang, 2021, [24]). Liu et al. [25] developed a phenology-based method to identify cropping patterns, but they did not consider any environmental variables as drivers. Wu et al. [26] proved that spatial patterns of cropping systems and phenology in Chinese cropland were highly related to the geophysical environmental factors, but without taking into account the impacts of biophysical forces and anthropogenic drivers. The knowledge of phenological patterns and their different drivers at landscape scale can provide a valuable support tool for planning agricultural land use conversion, managing climate change impacts, and developing adaptative strategies. The main impacts of climate change on agriculture include soil production decline, water security declines, and increasing frequency of weather extremes [27]; in this perspective, how landscape is organized and managed is central to achieving a balance between productive and other ecosystem services. A key challenge is therefore facing the lack of spatially differentiated management approaches since climate change causes regionally differentiated impacts [27]. To fill this research gap, in this study we propose a multivariate approach that is based on temporal NDVI profiles of crop types to quantify the dependency of crop seasonality on multiple environmental drivers, at landscape scale. The proposed method aims to identify crop seasonality using a three-year NDVI times series from Sentinel-2, to quantify the role of the main biophysical and land management controlling factors on the identified crop phenology patters, and to map such patterns.

## 2. Materials and Methods

### 2.1. Study Area

The study area is located in the Capitanata plain (Foggia province), the second largest plain in Italy (about 7000 km$^2$), in the northern part of Apulia region placed in the south-eastern part of Italy (Figure 1). The regional topography is mainly flat or slightly sloping, except for the Gargano area, situated in the northwest of the region. The climate of the study area is classified as Mediterranean semi-arid, characterized by moderately cold and rainy winters and dry summer seasons. Annual rainfall (avg. 550 mm/year) is unevenly distributed throughout the year, being mostly concentrated during the winter months. Long-term mean air temperature is 15.4 °C, while the average minimum and maximum yearly temperatures are 3.5 and 29.5 °C, respectively; however, temperatures may fall below 0 °C in winter and rise above 40 °C in summer. The soil texture of this area is predominantly and homogenously clay, apart from a southern sandy clay loam area [28].

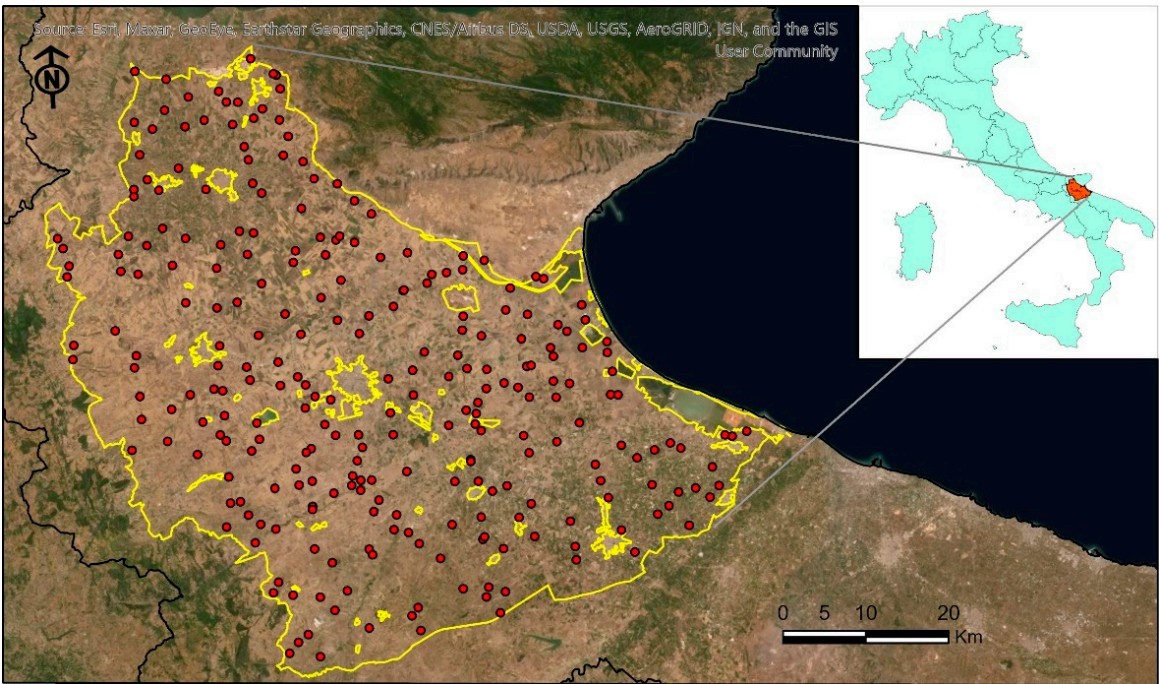

**Figure 1.** Location of the study area and distribution of the sample points. The red dots are the georeferenced sample points, and the yellow area represents the limit of the study area.

Due to its climatic conditions and land characteristics, Apulia is one of the most important regions in Italy for the agriculture production: in particular Foggia province is important for wheat, vegetables, and olive production, accounting for about 12%, 13%, and 4% of the wheat, total open air growing area, and the amount of olive surface at national level, respectively [29].

### 2.2. Environmental Data

In the present study the environmental data considered are soil variables (SOIL), climate information (CLIM), and land use (LU).

The soil profiles were extracted from the National Soil Database (NSD; https://esdac.jrc.ec.europa.eu/content/carta-dei-suoli-ditalia-soil-map-italy, accessed on 17 June 2021), complete of physical and chemical analytical data. Soil information gathered and harmonized in the NSD was collected from different soil survey projects: 169 soil mini-pits from the statistical monitoring program of the Italian Ministry of Agricultural and Forestry Policies, and 95 soil profiles from soil survey of Apulia Region at 1:50,000 scale. This set of

264 georeferenced sample points, distributed throughout the study area, were characterized with the biophysical variables described in the following sections.

The pedological (SOIL) variables selected are (Table 1): texture (sand and silt), soil organic carbon content (OCC), and available water content (AWC). We considered these variables because they are strictly connected with the crop growing. Soil texture indicates the relative content of particles of various sizes, such as sand, silt, and clay in the soil; it is an important soil characteristic that influences stormwater infiltration rates and consequently how much water is available to the plant [30]. It is one of the most important properties of a soil since it greatly affects crop production, land use, and management The OCC is important to soil nutrient status in agroecosystems, has an important role in the physical, chemical, and biological function of agricultural soils [30]. AWC is an indicator of a soil's ability to retain water and to make it sufficiently available for plant use. AWC is the water held in soil between its field capacity and permanent wilting point [31].

**Table 1.** Soil data statistics in the study area.

|  | Sand (%) | Silt (%) | OCC (Unitless) | AWC (Unitless) |
|---|---|---|---|---|
| Min | 0.5 | 0.5 | 0.14 | 59.07 |
| Max | 96.5 | 80 | 4.5 | 219.14 |
| Mean | 28.8 | 35.43 | 1.28 | 140.70 |

Climatic (CLIM) variables were acquired from the WorldClim V2 dataset (http://www.worldclim.org/bioclim, accessed on 17 June 2021) which is a set of 1970–2000 global climate layers (monthly gridded temperature and precipitation data) with a spatial resolution of 1 km$^2$ [32]. Even though the NDVI dataset refers to 2017–2019 and in 20 years there could have been evident variations in climate, their effects on crops may involve the associated responses (e.g., anticipation in the growing season, prolonged duration, enhancer the production), rather than the kind of influence itself. Furthermore, WorldClim V2 dataset represents climate annual trends, seasonality, and extreme or limiting factors, and consequently it has been widely used for agroecological studies. The climatic variables considered in this study were the following: Max Temperature of Warmest Month ($T_{max}$), Min Temperature of Coldest Month ($T_{min}$), Annual Precipitation ($P_{tot}$). For each soil sample point, the corresponding CLIM variables were extracted.

Land management information of the study area were derived from the Land Use (LU) map of Apulia region (www.sit.puglia.it, accessed on 17 June 2021). The LU map referred to year 2011 and was realized according to the Corine Land Cover (CLC) project (https://www.eea.europa.eu, accessed on 17 June 2021). Since agricultural land (class 2) in CLC database for Apulia region did not change much from 2012 and 2017—only 0.06% (from 15,615.7 km$^2$ in 2012 to 15,606.2 km$^2$ in 2017) [33], in this study the LU map was chosen for its relatively higher spatial resolution (1:5000) with respect to the CLC map, although it is older. For this study, only agricultural classes were selected (Table 2).

**Table 2.** Main agricultural land use classes of the Capitanata plain and their surface.

| LU Code | Description | Surface (ha) |
|---|---|---|
| 211 | Non irrigated arable land | 70,915 |
| 212 | Permanently irrigated lands | 181,107 |
| 221 | Vineyards | 28,563 |
| 223 | Olive groves | 28,858 |

*2.3. Satellite Data*

For this research, Copernicus Sentinel-2 (S2) imagery of the study area was collected for years 2017, 2018, and 2019. All available images were downloaded and analyzed through Google Earth Engine platform (https://code.earthengine.google.com, accessed on 17 June 2021) [34]. S2 imagery was collected with an approximate 5-day temporal

resolution. Band 4 (red, 10 m spatial resolution) and band 8 (near infrared, 10 m spatial resolution) were used to compute NDVI. Images featuring more than 5% of clouds and cirrus pixels were discarded. Cloudy pixels on remaining images were masked using the QA60 bit-mask band provided. The QA60 band masks opaque and cirrus clouds at 60 m spatial resolution. Due to its coarser resolution than the optical bands, NDVI may be computed on undetected cloudy pixels, particularly at the boundary between the cloud and non-cloud mask [35], resulting in out of bound values. NDVI profiles for the sample points were collected, and profiles consisting of less than 15 time points per year were discarded. To address the removal of entire cloudy images, the masking of cloudy pixels and the presence of out of bound NDVI values at the cloud's boundaries a harmonic model of time was fitted to each profile and NDVI harmonic trajectories were predicted every 15 days (Figure 2):

$$ND\hat{V}I_{t,x,y} = \alpha_{x,y} + \delta_{x,y}t + \sum_{\omega=1}^{3}\left[\beta_{\omega,x,y}cos(2\pi\omega t) + \gamma_{\omega,x,y}sin(2\pi\omega t)\right] \tag{1}$$

where $\alpha,\delta,\beta_\omega$, and $\gamma_\omega$ are harmonic model coefficients fitted to each $(x,y)$ coordinates pair. The resulting datasets consisted of 25 images per year (Table 3); for each sample point, the modeled temporal trajectories for the years 2017, 2018, and 2019 have been extracted.

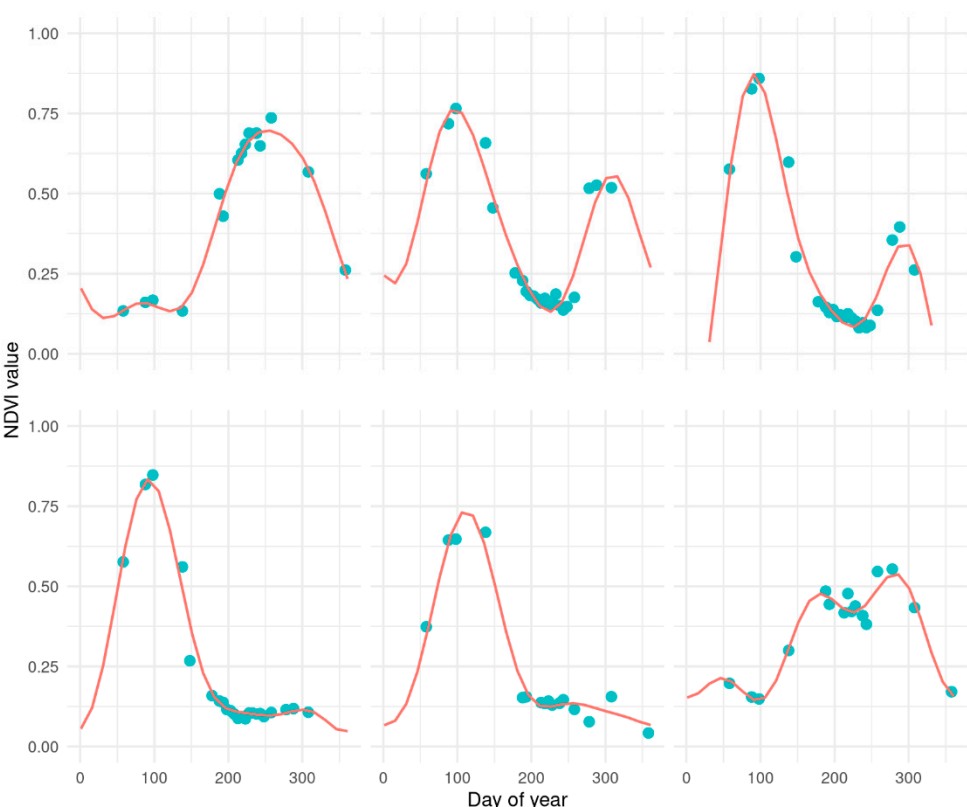

**Figure 2.** Six exemplary sample points of NDVI values collected for year 2017 (cyan dots) and harmonic models of time fitted to them (red lines). Missing NDVI values were due to cloudy image removals and clouds masking.

**Table 3.** Correspondence between NDVI temporal bands and date.

| Temporal Bands | Date | DOY | Temporal Bands | Date | DOY |
|---|---|---|---|---|---|
| b01 | 1-Jan | 1 | b14 | 15-Jul | 196 |
| b02 | 16-Jan | 16 | b15 | 30-Jul | 211 |
| b03 | 31-Jan | 31 | b16 | 14-Aug | 226 |
| b04 | 15-Feb | 46 | b17 | 29-Aug | 241 |
| b05 | 2-Mar | 61 | b18 | 13-Sep | 256 |
| b06 | 17-Mar | 76 | b19 | 28-Sep | 271 |
| b07 | 1-Apr | 91 | b20 | 13-Oct | 286 |
| b08 | 16-Apr | 106 | b21 | 28-Oct | 301 |
| b09 | 1-May | 121 | b22 | 12-Nov | 316 |
| b10 | 16-May | 136 | b23 | 27-Nov | 331 |
| b11 | 31-May | 151 | b24 | 12-Dec | 346 |
| b12 | 15-Jun | 166 | b25 | 27-Dec | 361 |
| b13 | 30-Jun | 181 | | | |

*2.4. Methodology*

The multivariate approach used in this work consists of three steps according to the flowchart of Figure 3.

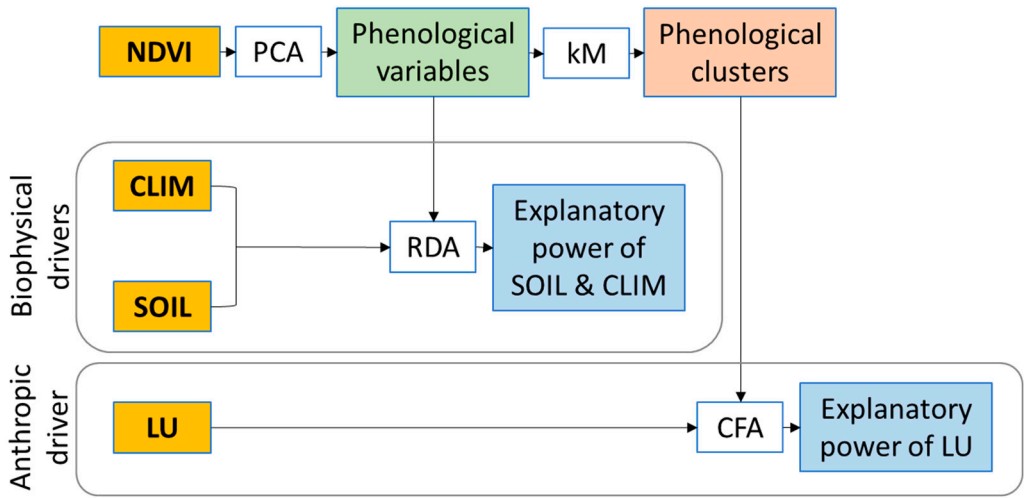

**Figure 3.** Flowchart of the multivariate approach used in this work.

First, information on the crop phenology of Capitanata was obtained from the S2 NDVI modeled time-series from 2017 to 2019 (25 images per year). On this basis, we performed a Principal Component Analysis (PCA) to summarize the phenological information associated to the temporal bands of the three annual NDVI time-series [14,36] and took into consideration the first three PCs.

In the second step, we performed a Redundancy Analysis (RDA) on the sample points to explore the explanatory power of the quantitative biophysical drivers on the crop phenology patterns through the years. RDA is a supervised multivariate statistical technique that measures redundancy, i.e., the proportion of the total variance of one set of response variables explained by a canonical variate from another set of explanatory variables [37]. Accordingly, the RDA axes represent the percentage of the variance of the response variables explained by the predictors. The first three PCA axis of each year were considered as response variables, while the bioclimate and soil variables were used as predictors. Due to their high heterogeneity, the explanatory variables were standardized prior to analysis.

Then, to identify homogeneous units in terms of phenology patterns (i.e., pheno-clusters, PhCls), a k-Means (kM) classification was performed all over the study area based

on the first three PCA axis of each year, and the mean NDVI annual profile for each cluster was computed. Finally, to compute the separation among the different pheno-clusters in terms of land use, we carried out a Corresponding Factor Analysis (CFA) across the whole study area. The CFA is a multivariate technique that detects associations and oppositions existing between categorical subjects (LU types) and objects (PhCls), measuring their contribution to the total inertia for each factor [38]. All the statistical analysis of this study were performed with XLSTAT [39].

## 3. Results

The PCA of the NDVI temporal bands provided high proportions of explained variance for the first three principal axes (PC1, PC2 and PC3), 80% for 2017, 79% for 2018, and 77% for 2019, respectively (Figure 4). According to Figure 5, all the three years behaved similarly: the first PC is mainly related to the summer NDVI temporal bands, while the second PC to the fall-winter bands, and the third PC to the spring ones. Accordingly, the first three PCs can be considered a synthetic expression of the crop annual seasonality in terms of timing and number of productivity peaks.

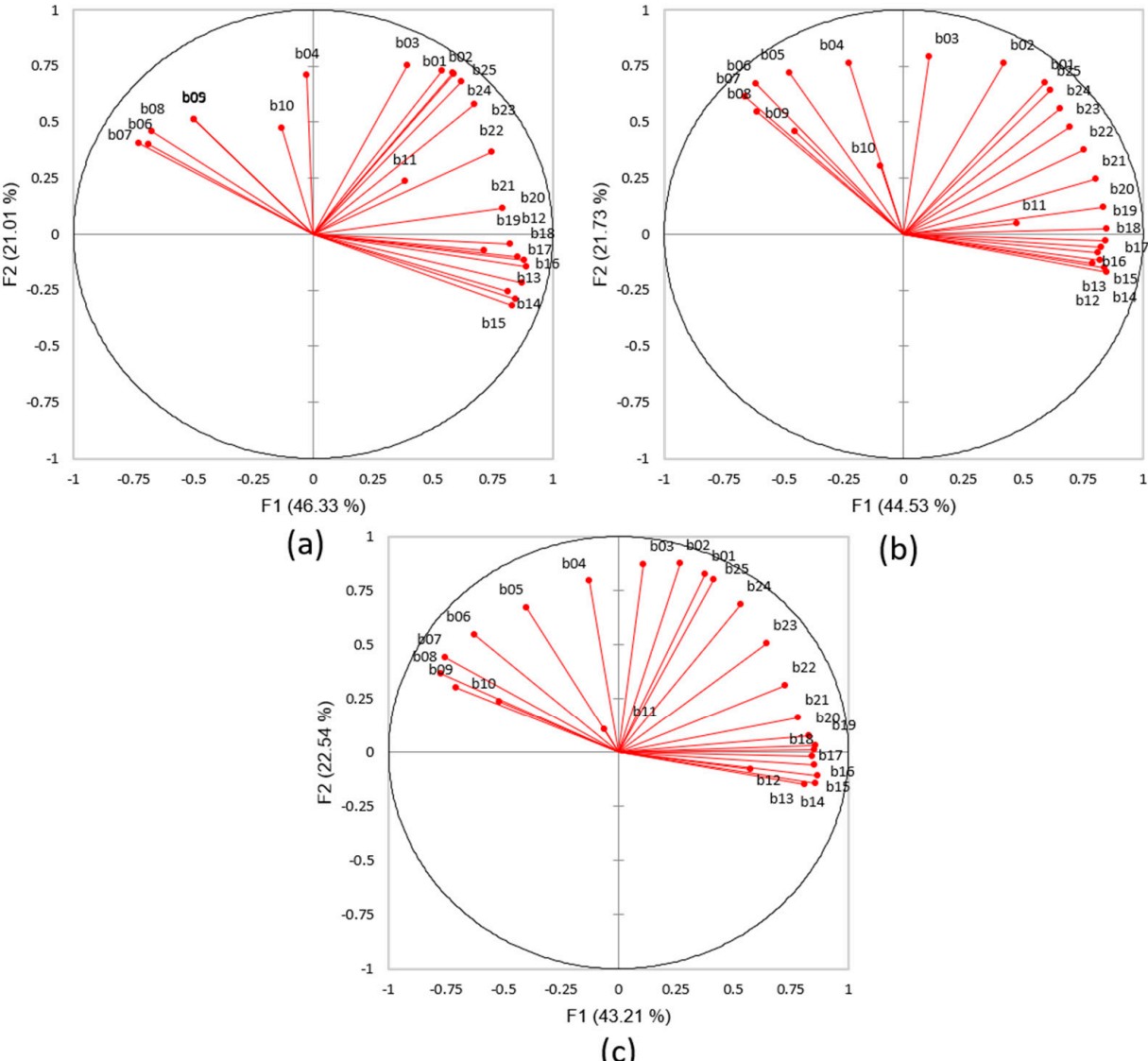

**Figure 4.** Principal Component Analysis (PCA) biplot for each year with the distribution of the 25 NDVI temporal bands (red vectors) according to the first two PCA axis: (**a**) 2017; (**b**) 2018; (**c**) 2019.

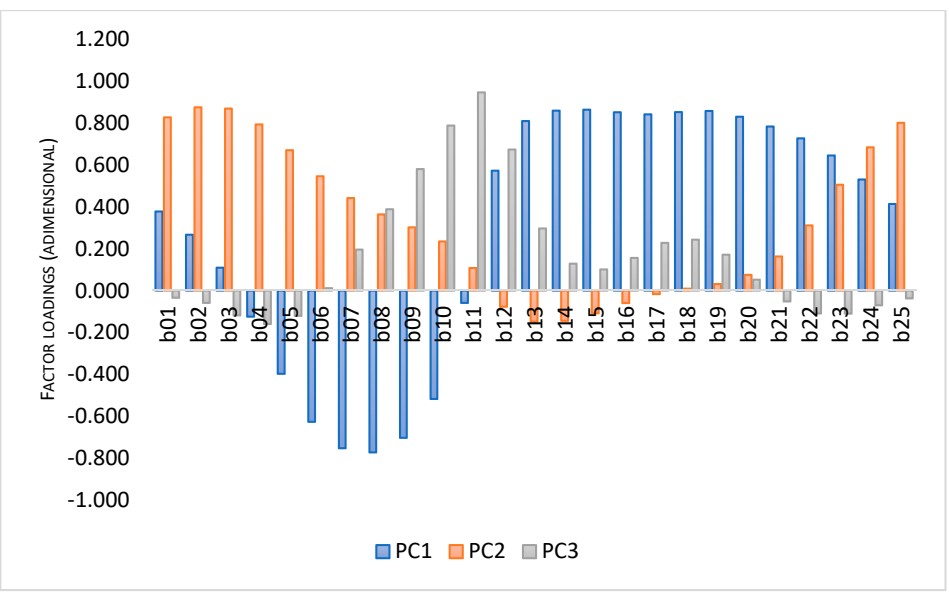

**Figure 5.** Histogram of the Principal Component Analysis (PCA) factor loadings of the NDVI temporal bands over the first three PC axis (year 2019 as an example).

The canonical axes obtained from the RDA between the first three PCs and the CLIM and SOIL driving variables (Figure 6) explained about 80% of the total variance of the response variables, with a high significance level ($p < 0.001$). Considering the coarse-scale approach of this study, this result proves the strong influence of soil and climate on the vegetation seasonality of Capitanata crops, and that for the three years considered the PCs have the same phenological meaning. RDA demonstrated that PC1 is guided by soil variables, while PC2 and PC3 by climatic ones. In detail, crops with summer productivity (i.e., PC1) are mainly driven by soils with high value of sand in the texture and low values of OCC. To the contrary, crops characterized by a fall-winter productivity (i.e., PC2) are controlled by high $T_{min}$, high variability in annual precipitations, and low AWC values. Meanwhile, PC3, linked to spring productivity, resulted as explained by low $T_{max}$, large amount of annual precipitation, and soils with high value of silt in the texture. This means that crops with high summer NDVI values, and thus with peak of productivity during the dry season, are usually associated to drained soils. Crops with high PC2 values, i.e., autumn-winter crops, mainly depend on mild winter times and a marked alternance of dry and wet period. Finally, crops with high spring NDVI values, and thus with a peak of productivity during the wet season, are controlled by abundant rainfall and high available water content.

The k-Means classification allowed distinguishment of four phenologically homogeneous clusters, called pheno-clusters (PhCls). Figure 7 shows their mean annual NDVI profile. PhCl1 showed highest NDVI values during the summer season and moderate values in the other months; this could represent the typical NDVI curve of permanent tree crops (i.e., vineyards, olive groves, and fruit trees). PhCl2 showed an opposite, bimodal behavior: low NDVI values during the dry season and high NDVI values during spring and fall-winter. PhCl2 is characterized by a rainfed crop NDVI curve, with homogeneous coverage: during the summer season NDVI values are low, approximately between 0.2 and 0.4, like for grasslands, and it could be associated with rainfed pastures, largely diffuse in the Capitanata.

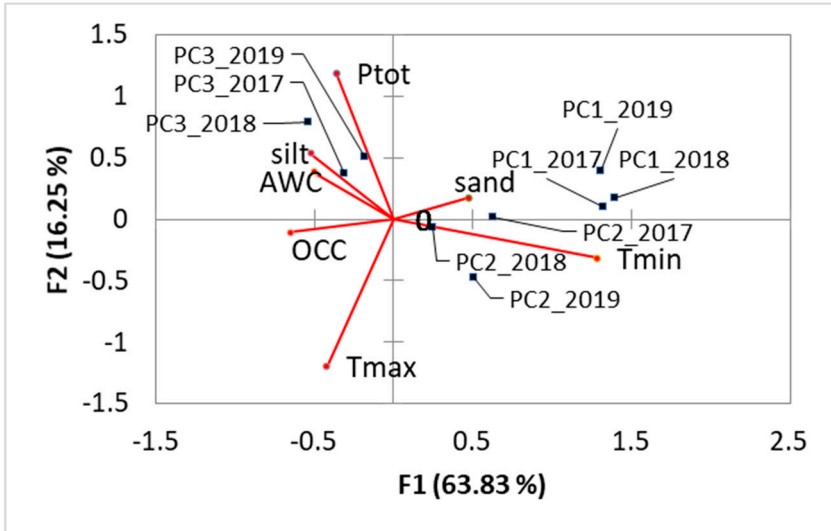

**Figure 6.** Correlation biplot based on Redundancy Analysis (RDA) performed in the sample points with the SOIL and CLIM variables as predictors, and the PCs for each year as response variables.

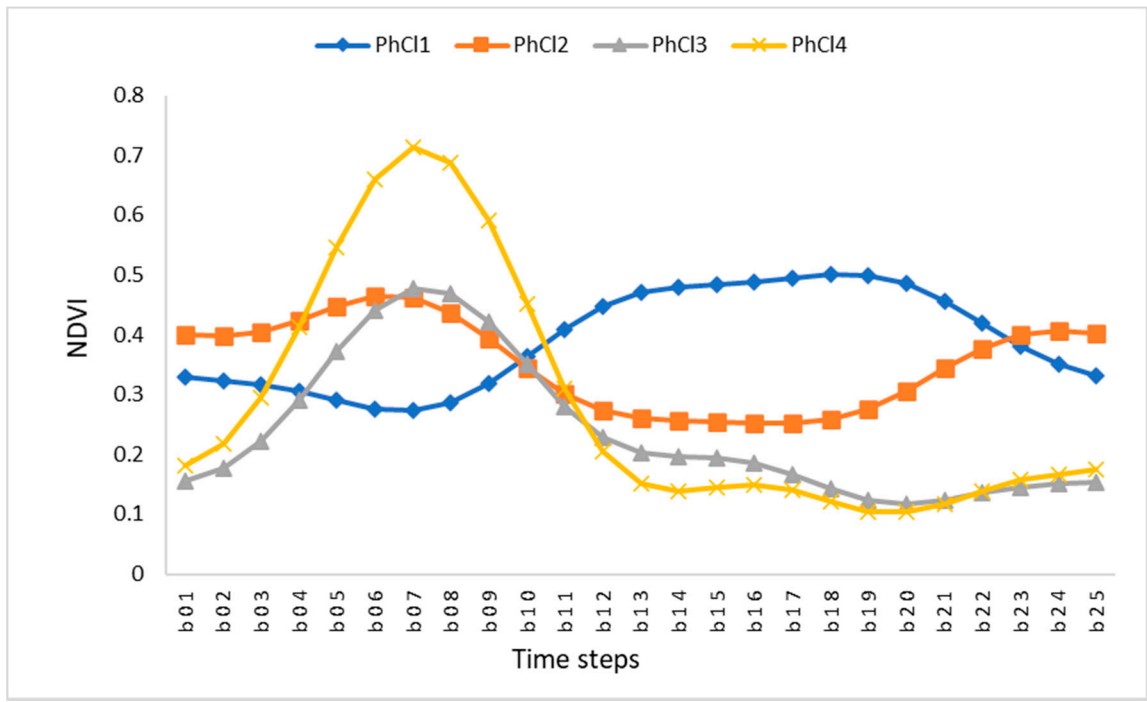

**Figure 7.** Mean NDVI profile of the four pheno-clusters (PhCls) identified.

PhCl3 and PhCl4 showed a marked unimodal NDVI curve with a peak in spring in both cases, but with different intensity, lower in the former (NDVI value around 0.5) and higher in the latter (NDVI value around 0.7). During summertime, NDVI presents very low values, less than 0.2, typical of bare soil, and these curves could be associated with rainfed arable land. The NDVI of PhCl4 has the typical curve related to a single winter cropping system, like winter wheat which is the dominant cultivation in this area.

The CFA biplot of Figure 8 (about 83% of explained variance) shows that the PhCl1 is positively correlated with the agricultural LU class of 221 and 223, while LU class 211 corresponds to two different pheno-clusters: PhCl2 and PhCl4, classes associated at typical NDVI related to single winter crops. The non-irrigated arable lands (211) resulted as mainly explained by high late spring NDVI-based productivity; the presence of this LU type is mainly linked to the water load derived from the early spring precipitations which nourish

the subsequent growing season. The irrigated arable lands (212) were mainly characterized by high NDVI values during spring-summer, and low NDVI values in winter; this LU type is the less dependent on the precipitation seasonality, due to the human-based water provision, and hence its NDVI-based productivity can continue also during the dry months. Finally, the LU classes of vineyards and olive groves (221 and 223, respectively) were explained by high values of NDVI-based productivity from summer to winter; this LU type represents a perennial tree cropping system, able to sustain the summer dryness and to maintain green leaves until fall-winter, when the photosynthetic activity gradually stops, or even beyond (e.g., olive groves are evergreen crops). Figure 9 shows the distribution of the pheno-clusters throughout the study area.

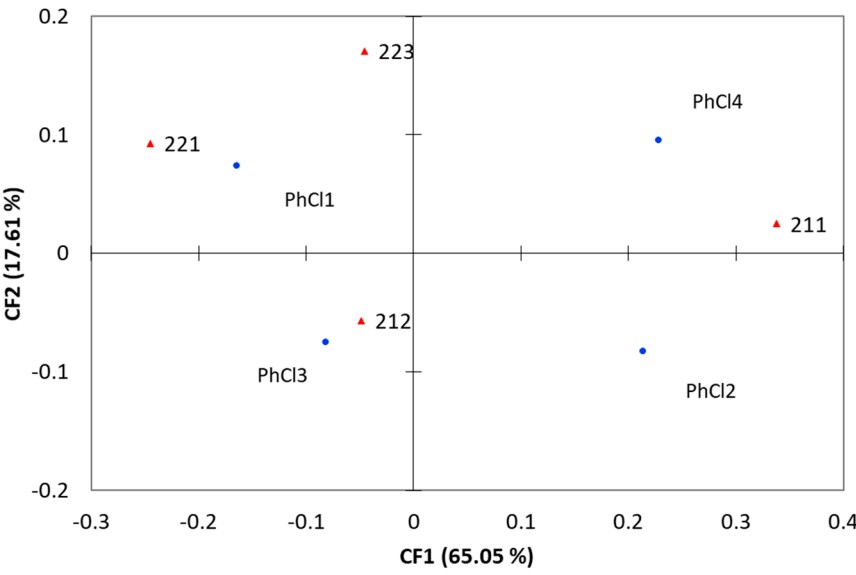

**Figure 8.** CFA biplot between the four pheno-clusters (PhCl) (blue dots) and the agricultural land use types (red triangles).

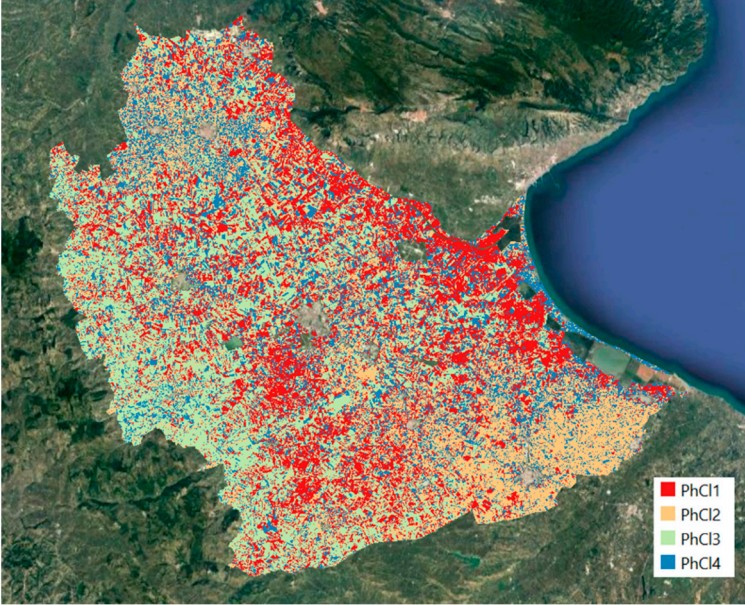

**Figure 9.** Distribution of the pheno-clusters throughout the study area.

## 4. Discussion

This paper attempted to identify crop phenology patterns in a robust and cost-efficient way across a large agricultural landscape in a Mediterranean region and potentially in

retrospective, by using a multivariate approach. In this study, the spatial and temporal variability of crop growth was assessed using remote sensing phenological information in relation to the main biophysical and anthropogenic drivers. Accordingly, our study demonstrates the potential of satellite-based phenology to provide information about temporal and spatial variability of crop growth across a typical Mediterranean landscape, which, additionally, may provide relevant information for agricultural management. Phenological patterns provide comprehensive insight of the spatio-temporal crop growth variability across the variable seasons, which advances our understanding of the crop responses to changing conditions, at local to regional scale [40]. The method proposed in this study provides a pathway towards effective estimation and monitoring of crop growth variability through time and space, which is a key concern for sustainable agriculture success. It can be used to develop future, more detailed studies to fully utilize the potential of phenological indicators for site characterization, monitoring, and prevision of the impact of extreme weather events, identification of crop response patterns to a disease, etc.

Results confirmed that climate, in terms of precipitation and air temperature, is the main driver of phenology in a Mediterranean landscape, in particular for arable crops, by controlling soil moisture and water availability to plants and affecting evapotranspiration [41]. Precipitation seasonality, high temperature, and consecutive droughts in this region strongly affect the crop cover dynamics and have resulted in adaptations of farming system management in response to climatic variation. These factors could all have affected the phenological variability and the productivity of the cropping systems in the study region, resulting in phenological changes over space. The results from this study are expected to represent a framework for other investigations about agriculture adaptation and mitigation strategies, for instance, to drought and water stress. The phenology datasets and the trend results could be combined with climate data to estimate the crop water requirements and provide a tool for landscape managers and stakeholders to make decisions for the extension of agricultural areas according to the available water resources in a context of water stress. In this study, we simplified anthropogenic factors into generic land use, therefore, further work is needed to separate the effects of climate and human activities fully and precisely on agroecosystems at local scale [23].

Furthermore, the framework presented allowed to map the crop phenology pattern distribution across the landscape. Due to the dynamic character of agricultural systems, crop mapping based on multi-temporal approaches is superior to single-date image analyses [19]. While traditional approaches using classification algorithms entail field observations to train or test the classifier, the use of crop-specific VI temporal profiles (i.e., behavior of a certain crop type throughout the year) is independent of ground truth data [19]. Several studies investigated the use of crop-specific seasonal profiles for crop discrimination and mapping at different spatial scales, from local to regional level [18,19]. VI temporal features are taken as the major theoretical basis for distinguishing crops from other vegetation, and one crop type phenology from another [41]. Time-series-based methods for cropping pattern identification exploit the fact that VI annual trends representing a specific crop seasonality are usually more similar than profiles representing different crops [19,42].

## 5. Conclusions

Our analysis demonstrates the potential of phenology to assess crop growth variability and to provide a comprehensive understanding of the joint role of soil, climate, and land use on crop seasonality. Single vegetation index images have been successfully utilized to recognize crop variability across a region [43,44]. However, the single image approach has been criticized for lacking information on intra-seasonal growth dynamics [45]. Multi-seasonal images and the associated phenological patterns allow the revelation of the intra-annual biophysical properties of crops across the landscape, as jointly driven by soil, climate, and land use [40]. This potentially provides a better understanding of crop variability, which is a key factor to improve management practices at farm level and to monitor land use changes in agricultural areas.

Furthermore, the method presented in this analysis shows the efficacy of phenology to recognize crop growth variability, obtained in a cost-effective way, over large areas, using high resolution satellite. Considering the increasing availability of remote sensing imagery, the spatio-temporal variability estimation using phenological patterns can provide valuable information for agriculture suitability assessment, in terms of energy demand and water stress. Currently, Sentinel-2 is the appropriate imagery for such analysis thanks to its high spatial and temporal resolution, suitable to study heterogeneous landscapes, like agricultural ones, and seasonal phenomena, like those related to crop phenology.

**Author Contributions:** Conceptualization, S.B.; methodology, S.B. and S.V.; formal analysis, S.B. and M.B.; investigation, S.B. and S.V.; data curation, S.V.; writing—original draft preparation, S.B.; writing—review and editing, S.V., M.B. and R.N.; project administration, S.B. All authors have read and agreed to the published version of the manuscript.

**Funding:** This research was funded by Italian Ministry of Agriculture, AgriDigit program (DM 36503.7305.2018 of 20 December 2018).

**Institutional Review Board Statement:** Not applicable.

**Informed Consent Statement:** Not applicable.

**Data Availability Statement:** The data came from the Council for Agricultural Research and Economics, Research Centre for Agriculture and Environment (CREA-AA).

**Conflicts of Interest:** The authors declare no conflict of interest. The funders had no role in the design of the study; in the collection, analyses, or interpretation of data; in the writing of the manuscript; or in the decision to publish the results.

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
