# Peer review of "Exploring the Drivers of Sentinel-2-Derived Crop Phenology: The Joint Role of Climate, Soil, and Land Use"

_land, doi:10.3390/land10060656_

Round 1

Reviewer 1 Report

Dear all,

Thank you for the opportunity to read this interesting paper.

The use of satellites to investigate the state of vegetation in space and time is now a widely used and scientifically strong approach. Specifically, here the spatial and temporal variability of crop growth was assessed using remote sensing phenological information in relation to the main biophysical and anthropogenic drivers.

The analysis is correctly conducted, and the paper is well structured. Results are interesting and they are well discussed.

The below aspects should be addressed by the author before publication.

SPECIFIC COMMENTS:

Line 101 – 104: Could you cite the source of these data?

Figure 1: Could you report the scale on the map?

Line 273: “within the AGRIDIGIT project” - since this is the first time it has been mentioned, either explain it briefly or put it only in the funding section.

I would briefly discuss how the result can be a support for agricultural management.

Author Response

Response to referee 1

Dear all, Thank you for the opportunity to read this interesting paper.

The use of satellites to investigate the state of vegetation in space and time is now a widely used and scientifically strong approach. Specifically, here the spatial and temporal variability of crop growth was assessed using remote sensing phenological information in relation to the main biophysical and anthropogenic drivers.

The analysis is correctly conducted, and the paper is well structured. Results are interesting and they are well discussed.

We thank you for this generally positive evaluation of our study.

The below aspects should be addressed by the author before publication.

SPECIFIC COMMENTS:

Line 101 – 104: Could you cite the source of these data?

Done, we included reference.

Figure 1: Could you report the scale on the map?

Done

Line 273: “within the AGRIDIGIT project” - since this is the first time it has been mentioned, either explain it briefly or put it only in the funding section.

Done, we include it in the funding section.

I would briefly discuss how the result can be a support for agricultural management.

We thank the reviewer for the hint. We added a line about this in the Conclusion section (L372 Pag13).

Reviewer 2 Report

This work aims to analyze the relationship between crop phenology and environmental variables. The topic is interesting, although widely studied already. However, the current manuscript lacks massive details and justifications. The significance of the study is not well justified.

Major concerns:

  • The literature review is not enough.
  • The methods were not well justified.
  • Some important details about the datasets are missing.
  • Many conclusions are not closely related to or not supported by the study.

Specifically:

L81: Firstly, simply search crop phenology driver in Google Scholar would give you thousands of results. I wonder what the authors mean by “in agricultural studies this kind of research question has received far less attention”. Secondly, what has been studied so far about relating environmental factors to crop phenology patterns?

Figure 1.

  • What do the yellow polygon and red points stand for? Please add a legend or explain in the figure caption.
  • What samples are there?
  • What are the numbers on the horizontal and vertical axes?
  • Please add north arrow and scale bar.
  • What is the base map? Consider showing the LU map as the base map.

L119: Please define the four pedological variables and add reference(s).

L127: Was this dataset created by interpolation? What was the reported accuracy? Please provide a brief introduction.

L128: Why use years 1970-2000 to represent the climate? Since your satellite data are for years 2017-2019, the near 20 years difference could mean big change in the climatic variables. Are there other climate datasets available? If not, quantified climate change information from the literature may be borrowed to make up for the change. In addition, the potential bias in your result and conclusion should be discussed.

L135: Please note that ‘5.000’ is the same as 5.

L136: Here you have another very strong assumption that the land use did not change from 2011 to 2017. I doubt that can be true, especially for agricultural lands. Please justify.

L138: By “geometric resolution”, do you mean spatial resolution? What was the spatial resolution of the LU map (how many meters was the pixel size)?

L139: What does CLC stand for?

Do you mean the LU map you selected has higher spatial resolution than the CLC map?

Table 2: What are the reported accuracy of the LU map, and for each of the four classes?

L147: How was the QA60 band used?

L148: What do you mean by “from further elaboration”?

L150: The complete form of NDVI has already appeared in line 57. Write NDVI directly here.

L156: Please check this sentence. The grammar is wrong. What is it that you want to express?

Figure 3:

  • Why would the PCA of NDVI lead to the phenological variables?
  • Why did you use two different methods (RDA and CFA) for the analyses of the three types of variables (CLIM, SOIL, and LU)?

L173: After PCA, what was used to represent the phenological information?

L181: Why the first three PCs?

L185: homogeneous should be homogeneity.

L187: What do you mean by “the resulting clusters were characterized according to the shape and productivity peaks of their mean annual NDVI profiles”? What does “characterized” mean here?

L197: Still, over 20% information was discarded. Is the first three PCs enough? Was the same information retained in the three different years?

Figure 4: The labels of the NDVI bands are not legible. Please re-create the figure.

L198 and Figure 5:

  • It is not clear how each PC corresponds to the seasons. Please add explanations. For example, you can divide the horizontal axis to four different sections and mark the seasons accordingly.
  • You used 2019 as an example, but how representative is it?
  • Figure 5: Add label for the vertical axis.

L228: How was the mean annual NDVI profile calculated? Did you pick one pixel and computed the mean NDVI of this pixel over the three years, or did you use multiple pixels?

Figure 7.

  • Why is the horizontal axis called time steps?
  • Decide to represent the NDVI bands by uppercase or lowercase symbols. It should be unified throughout the manuscript.
  • Besides the mean value, could you also show the deviation (e.g., by showing multiple lines or adding error bars)?

Figure 8:

  • CF or FC?
  • Besides different colors, consider to also use different symbols, for easier figure reading.

Figure 9:

  • Please use more distinctive colors.
  • Since the clusters are associated to the land use types, it is helpful to show both maps. The LU map can be shown in Figure 1

L273: What does AGRIDIGIT represent?

L282: Better than what? Why?

L324: Sure multi-temporal data can potentially help understand the temporal variability of crops. But I don’t see how that (especially the intra-annual variability) is reflected by your study.

L330: How can you get the conclusion that Sentinel-2 is the most appropriate?

Author Response

Response to referee 2

This work aims to analyze the relationship between crop phenology and environmental variables. The topic is interesting, although widely studied already. However, the current manuscript lacks massive details and justifications. The significance of the study is not well justified.

Major concerns:

  • The literature review is not enough.
  • The methods were not well justified.
  • Some important details about the datasets are missing.
  • Many conclusions are not closely related to or not supported by the study.

Thanks for the suggestions: we added some references in the Introduction section, we specified in a better way the methodology and the dataset and we improved conclusions.

 Specifically:

L81: Firstly, simply search crop phenology driver in Google Scholar would give you thousands of results. I wonder what the authors mean by “in agricultural studies this kind of research question has received far less attention”. Secondly, what has been studied so far about relating environmental factors to crop phenology patterns?

We agreed with the referee and amended the paragraph accordingly, by highlighting the scale of analysis at which there is this research gap, i.e. the landscape. We also added a brief related state-of-the-art, as suggested.

 Figure 1.

  • What do the yellow polygon and red points stand for? Please add a legend or explain in the figure caption.
  • What samples are there?
  • What are the numbers on the horizontal and vertical axes?
  • Please add north arrow and scale bar.
  • What is the base map? Consider showing the LU map as the base map.

We added all information for a more comprehensive figure.

 L119: Please define the four pedological variables and add reference(s).

I see referee's point; we added a sentence about the pedological variables (sand and clay are in soil texture) and the related reference.

 L127: Was this dataset created by interpolation? What was the reported accuracy? Please provide a brief introduction.

The pedological information are the result of a field campaign; so they are related to each sample point and are not interpolated.

L128: Why use years 1970-2000 to represent the climate? Since your satellite data are for years 2017-2019, the near 20 years difference could mean big change in the climatic variables. Are there other climate datasets available? If not, quantified climate change information from the literature may be borrowed to make up for the change. In addition, the potential bias in your result and conclusion should be discussed.

Thank you for highlighting this point. The used Worldclim dataset is the most updated one, and we chose it because it has agroecological meaning, it is reliable, well-known and at an adequate spatial resolution. We agree that in 20 years there could have been evident changes in climate, but we do believe that the influence of climate on basic phenology is the same: what could have been changed is the trend in the response (i.e. anticipated growing season, prolonged duration), not the "sign" of the influence.  We tried to clarify this also in the text.

 L135: Please note that ‘5.000’ is the same as 5.

Done, we corrected the error.

L136: Here you have another very strong assumption that the land use did not change from 2011 to 2017. I doubt that can be true, especially for agricultural lands. Please justify.

We see referee's point. Yet, since according to 6-ISPRA, 2018, agricultural land (class 2) in CLC database for Apulia region didn’t change so much from 2012 and 2017 (from 15615.7 km2 in 2012 to 15606.2 km2 in 2017), we decided to use a higher spatial resolution dataset, even if older.

L138: By “geometric resolution”, do you mean spatial resolution? What was the spatial resolution of the LU map (how many meters was the pixel size)?

We changed in the text.

L139: What does CLC stand for?

Done, we insert in the text.

Do you mean the LU map you selected has higher spatial resolution than the CLC map?

Yes, we corrected in the text.

Table 2: What are the reported accuracy of the LU map, and for each of the four classes?

L147: How was the QA60 band used?

Thanks for this remark. We acknowledge that cloud masking description was unclear. QA60 band is provided as a cloud mask band at 60 m resolution. Optical bands were masked with QA60 to remove (mask) cloudy pixels. We clarified the process in the text.

L148: What do you mean by “from further elaboration”?

We have rewritten the complete sentence to better explain the process of cloud-masking and image removal.

L150: The complete form of NDVI has already appeared in line 57. Write NDVI directly here.

Thanks for this suggestion. We removed the complete form.

 L156: Please check this sentence. The grammar is wrong. What is it that you want to express?

We have rewritten the entire sentence and moved it in the context of the cloud-masking process.

 Figure 3:

  • Why would the PCA of NDVI lead to the phenological variables?
  • Why did you use two different methods (RDA and CFA) for the analyses of the three types of variables (CLIM, SOIL, and LU)?
  •  

Because we are using, as input variables, the temporal bands; therefore, the first axis summarize the period of higher NDVI-based productivity. Please, in addition to the reference already cited in the text, refer to the following literature for further details:

Bellón, B.; Bégué, A.; Lo Seen, D.; de Almeida, C.; Simões, M. A Remote Sensing Approach for Regional-Scale Mapping of Agricultural Land-Use Systems Based on NDVI Time Series. Remote Sensing 2017, 9, 600, doi:10.3390/rs9060600.

  • Pesaresi, S.; Mancini, A.; Quattrini, G.; Casavecchia, S. Mapping Mediterranean Forest Plant Associations and Habitats with Functional Principal Component Analysis Using Landsat 8 NDVI Time Series. Remote Sensing 2020, 12, 1132, doi:3390/rs12071132.
  • de Almeida, T.I.R.; Penatti, N.C.; Ferreira, L.G.; Arantes, A.E.; do Amaral, C.H. Principal Component Analysis Applied to a Time Series of MODIS Images: The Spatio-Temporal Variability of the Pantanal Wetland, Brazil. Wetlands Ecology and Management 2015, 23, 737–748, doi:1007/s11273-015-9416-4.

We used two methods, because CLIM and SOIL are quantitative variables, while LU are categorical variables.

 L173: After PCA, what was used to represent the phenological information?

We used the first 3 Principal Components. We clarified this in the text.

 L181: Why the first three PCs?

Because together they explain 80, 79 and 77% of the variance, respectively in 2017, 2018 and 2019 (see Results section). Adding further axis would add only little information, and the scope of PCA is to reduce and summarize the study variables.

 L185: homogeneous should be homogeneity.

Done

L187: What do you mean by “the resulting clusters were characterized according to the shape and productivity peaks of their mean annual NDVI profiles”? What does “characterized” mean here?

We deleted the sentence because it did not describe any substantial analysis.

L197: Still, over 20% information was discarded. Is the first three PCs enough? Was the same information retained in the three different years?

The aim of PCA is to reduce the number of variables and summarize the information in few synthetic variables (xxx). 80% is a very good quota of information for biological systems (xxx); furthermore, the following PCs did not add large increase in the explained variance percentage. As written, the same information is retained in the three different years (80, 79 and 77%, respectively for 2017, 2018 and 2019).

 Figure 4: The labels of the NDVI bands are not legible. Please re-create the figure.

We see referee's point, and we modified Figure 4 in three separated biplots.

 L198 and Figure 5:

  • It is not clear how each PC corresponds to the seasons. Please add explanations. For example, you can divide the horizontal axis to four different sections and mark the seasons accordingly.
  • You used 2019 as an example, but how representative is it?
  • Figure 5: Add label for the vertical axis.

Thank you for pointing this out. Please, refer to Table 3 for the identification of the calendar day associated to each composite/temporal band and, consequently, of the corresponding season. In order not to overcharge the Figure, we did not add the information about the seasons; however, we followed referee's suggestion and added the y label. The representativeness of 2019 is witnessed by the biplots of Figure 4.

 L228: How was the mean annual NDVI profile calculated? Did you pick one pixel and computed the mean NDVI of this pixel over the three years, or did you use multiple pixels?

We computed the mean NDVI profile for each pheno-cluster (which is composed by multiple pixels) by averaging the NDVI value of each Sentinel composite over the three years.

 Figure 7.

  • Why is the horizontal axis called time steps?
  • Decide to represent the NDVI bands by uppercase or lowercase symbols. It should be unified throughout the manuscript.
  • Besides the mean value, could you also show the deviation (e.g., by showing multiple lines or adding error bars)?

We called it "time steps" because they are actually temporal bands (SENTINEL composites). We unified the symbols. We did not add the error bars because this is just a descriptive graph for characterizing the pheno-clusters, it has no statistical meaning; providing the error bars is out of the scope of the Figure.

 Figure 8:

  • CF or FC?
  • Besides different colors, consider to also use different symbols, for easier figure reading.

The referee is right, it is CF. Corrected and modified.

 Figure 9:

  • Please use more distinctive colors.
  • Since the clusters are associated to the land use types, it is helpful to show both maps. The LU map can be shown in Figure 1

We followed the referee's suggestion and updated Figure 1. Given the new arrangement of all the Figures according to both referees, we preferred not to add the LU map as a figure. Please, consider that to test the association, we performed a statistical analysis, so a visual comparison is redundant.

 L273: What does AGRIDIGIT represent?

Thanks for this remark, we included it in the funding section.

 L282: Better than what? Why?

We see referee's point and remove "better".

 L324: Sure multi-temporal data can potentially help understand the temporal variability of crops. But I don’t see how that (especially the intra-annual variability) is reflected by your study.

Thanks for this remark, we specified better in the text.

L330: How can you get the conclusion that Sentinel-2 is the most appropriate?

Because, as we stated, it has a high spatial and temporal resolution, suitable to study heterogeneous landscapes like the agricultural ones, and seasonal phenomena, like those related to crop phenology. We specified this in the text.

We would like to thank you for these helpful and constructive comments, which helped us to refine and further improve our analysis.

Reviewer 3 Report

Summary:

This study focuses on quantifying the dependency of crop seasonality on different environmental factors. For the said purpose, this study uses a three-year NDVI time series from Sentinal-2 and employs the multivariate approach using PCA (principal component analysis) and k-means classification approach.

Comments:

Please consider editing the abstract by adding one or two sentences related to problem statement and goal of the study.

Figure 4 , 5 & 6: The numbers on figures are overlapping. Please consider editing the figure to make it clear.

Please consider making a separate section for conclusion to make conclude the findings of the study.

Proofreading is required to improve the English language and remove the spelling and grammar errors.

Author Response

Response to referee 3

Comments and Suggestions for Authors

Summary:

This study focuses on quantifying the dependency of crop seasonality on different environmental factors. For the said purpose, this study uses a three-year NDVI time series from Sentinal-2 and employs the multivariate approach using PCA (principal component analysis) and k-means classification approach.

Comments:

Please consider editing the abstract by adding one or two sentences related to problem statement and goal of the study.

Thanks for the suggestions, we added some sentences.

Figure 4, 5 & 6: The numbers on figures are overlapping. Please consider editing the figure to make it clear.

We did not understand the comment; we do not see any overlapping in figure 5 and 6. We modified figure 4.

Please consider making a separate section for conclusion to make conclude the findings of the study.

Proofreading is required to improve the English language and remove the spelling and grammar errors.

We followed referee's suggestion: we separated Discussion from Conclusions and improved the English language.

We would like to thank you for these helpful and constructive comments, which helped us to refine and further improve our analysis.

Round 2
